**RESEARCH**                                                                                          **Open Access**

# Systematic assessment of gene co-regulation within chromatin domains determines differentially active domains across human cancers

Marie Zufferey[1,2,3], Yuanlong Liu[1,2,3], Daniele Tavernari[1,2,3], Marco Mina[1,2,3] and Giovanni Ciriello[1,2,3*]

\* Correspondence: giovanni.ciriello@unil.ch
[1]Department of Computational Biology, University of Lausanne (UNIL), Lausanne, Switzerland
[2]Swiss Cancer Center Leman, Lausanne, Switzerland
Full list of author information is available at the end of the article

## Abstract

**Background:** Spatial interactions and insulation of chromatin regions are associated with transcriptional regulation. Domains of frequent chromatin contacts are proposed as functional units, favoring and delimiting gene regulatory interactions. However, contrasting evidence supports the association between chromatin domains and transcription.

**Result:** Here, we assess gene co-regulation in chromatin domains across multiple human cancers, which exhibit great transcriptional heterogeneity. Across all datasets, gene co-regulation is observed only within a small yet significant number of chromatin domains. We design an algorithmic approach to identify differentially active domains (DADo) between two conditions and show that these provide complementary information to differentially expressed genes. Domains comprising co-regulated genes are enriched in the less active B sub-compartments and for genes with similar function. Notably, differential activation of chromatin domains is not associated with major changes of domain boundaries, but rather with changes of sub-compartments and intra-domain contacts.

**Conclusion:** Overall, gene co-regulation is observed only in a minority of chromatin domains, whose systematic identification will help unravel the relationship between chromatin structure and transcription.

**Keywords:** Chromatin compartment domains, Hi-C, Gene co-regulation

## Background

Chromosome conformation capture technologies have allowed exploring the three-dimensional (3D) organization of the chromatin in the nucleus. These approaches allow to quantify with what frequency two DNA loci are found in spatial proximity, independently of their contiguity along the genome sequence. In particular, high-throughput chromosome conformation capture (Hi-C) generates genome-wide maps of DNA contacts [1]. Computational analyses of these contact maps revealed

structural features of genome organization, among which (sub-)megabase chromatin domains characterized by frequent interactions within the domain and sparse interactions among different domains [2, 3] and large-scale compartments and sub-compartments [1, 4].

The formation of chromatin domains has been mostly attributed to two major mechanisms: chromatin loop extrusion mediated by cohesin and CTCF, and interactions among regions decorated by the same histone post-translational modifications [5–7]. Loop extrusion mediated by the cohesin complex is deemed responsible for the formation of structural loops delimited by CTCF binding, and domains associated with these loops have been called topologically associating domains (TADs). Different chromatin compaction and histone modifications are associated instead with the segregation of the chromatin in major compartments, in particular one characterized by high transcriptional activity (A compartment) and one enriched for heterochromatin and transcriptionally silenced regions (B compartment) [4, 8, 9]. However, chromatin epigenetic states encompass more than two states and recent studies have proposed different numbers of sub-compartments that better captured chromatin epigenetic features [4, 10]. Stretches of DNA assigned to the same compartment or sub-compartment have been termed compartment domains [10–12]. Notably, although conceptually distinct, TADs and compartment domains often overlap and/or have coincident boundaries [13]. Importantly, both compartment domains and TADs have been shown to be preferentially enriched for either active or inactive histone marks [2, 4, 11], and regulatory interactions such as those between enhancers and gene promoters typically occur within a domain rather than across different domains [14, 15]. Hence, it has been proposed that these structural elements can act as functional units.

The relationship between chromatin domains and transcriptional activity is however debated, and conflicting evidence has so far been reported. Disruption of domain boundaries has been shown sufficient to generate spurious enhancer-promoter interactions resulting in mis-regulated gene expression [16]. In particular, aberrant regulatory interactions have been investigated in cancer in association with somatic mutations altering the cell epigenome [17, 18] or chromosomal copy number changes leading to enhancer hijacking [19]. Even in the absence of altered domain boundaries, altered histone modifications within chromatin domains can affect regulatory interactions [20]. At the same time, genome-wide loss of structural loops induced by genetic experiments deleting CTCF [6] or cohesin [5] did not drive substantial transcriptional changes, casting doubts on the relevance of these loops in gene regulation. In a similar vein, computational analyses coupling gene expression and chromatin structure data reported in multiple instances higher co-regulation among genes within the same domain than among genes separated by a domain boundary [3, 17]. However, recent studies highlighted lack of concordance between domains of co-expressed genes and chromatin domains [21] or between chromatin contacts and gene expression [22].

These inconsistent results might arise from several factors. First, the association between gene expression and chromatin domains might be context dependent, e.g., more evident for specific genes under a certain condition rather than a genome-wide phenomenon. Second, Hi-C experiments are only available for a limited number of cell types and, often, previous studies have used a reference model, whose domains are unlikely to be universally conserved across tissues and species. Lastly, although genetic

experiments offer the opportunity of measuring chromatin and expression changes in a controlled system, these inevitably generate artificial conditions, which might be insufficient to simulate and stimulate the transcriptional diversity observed among cell types and cell states.

To overcome these challenges, here we analyzed the association between gene co-regulation and chromatin domains in a wide variety of conditions with tissue-matched Hi-C experiments. We developed an algorithmic approach to (1) test under which conditions gene co-regulation is associated with chromatin domains and (2) extract domains exhibiting significant differential activity between two conditions. In particular, we focused on comparing transcriptional activity between normal and cancer samples and among cancer subtypes. These datasets allowed us to analyze large-scale cohorts characterized by great transcriptional heterogeneity, which can be linked to specific molecular alterations and disease manifestations. We integrated gene expression data from multiple human cancer cohorts and Hi-C data from normal and cancer cell lines derived from the corresponding tissues. Our results consistently showed that gene co-regulation occurs only in a small, yet significant fraction of chromatin domains. These domains were enriched for less efficiently transcribed genes in the B sub-compartments and members of the same gene family. Importantly, by comparing Hi-C datasets from normal and tumor cells, we found that differentially active domains frequently change sub-compartment and exhibited intra-domain contact differences. Moreover, these domains provided complementary information to standard differential gene expression analyses. Hence, we expect that differentially active domains alongside differentially expressed genes will provide a more complete picture of transcriptional differences emerging in multiple biological contexts.

## Results

### The DADo algorithm

To systematically explore the extent of gene co-expression within chromatin domains, we formulated the problem as a comparison between two conditions and assessed gene expression differences between these two conditions within each domain (Fig. 1a). Domains comprising co-regulated and differentially expressed genes are said to be *differentially active*. Precisely, we designed a computational approach addressing two key questions: (1) are gene expression changes between two conditions more concordant within chromatin domains than expected? If so, (2) which domains exhibit significant evidence of differential activity between the two conditions?

To address the first question, we developed a score quantifying the concordance of gene expression fold-changes observed for genes within the same domain, accounting for both the sign and magnitude of the fold-changes. The fold-change concordance score (FCC) of a domain $t$ comprising $n$ genes $\mathbf{g} = \{g_1, ..., g_n\}$ is formally defined as:

$$\text{FCC}(t) = \left[\frac{2}{n}\sum_{\mathbf{g}}\delta\left(\frac{LFC(g_i)}{|LFC(g_i)|}, -1\right) - 1\right] * \left[2 \cdot \frac{\sum_{\mathbf{g}}|LFC(g_i)|\,\delta\left(\frac{LFC(g_i)}{|LFC(g_i)|}, -1\right)}{\sum_{\mathbf{g}}|LFC(g_i)|} - 1\right]$$

where $LFC(g)$ is the logarithm in base 2 of the mRNA expression fold-change of gene $g$, and $\delta(i, j)$ is the Kronecker delta function, which is equal to 1 if $(i = j)$, and 0 otherwise.

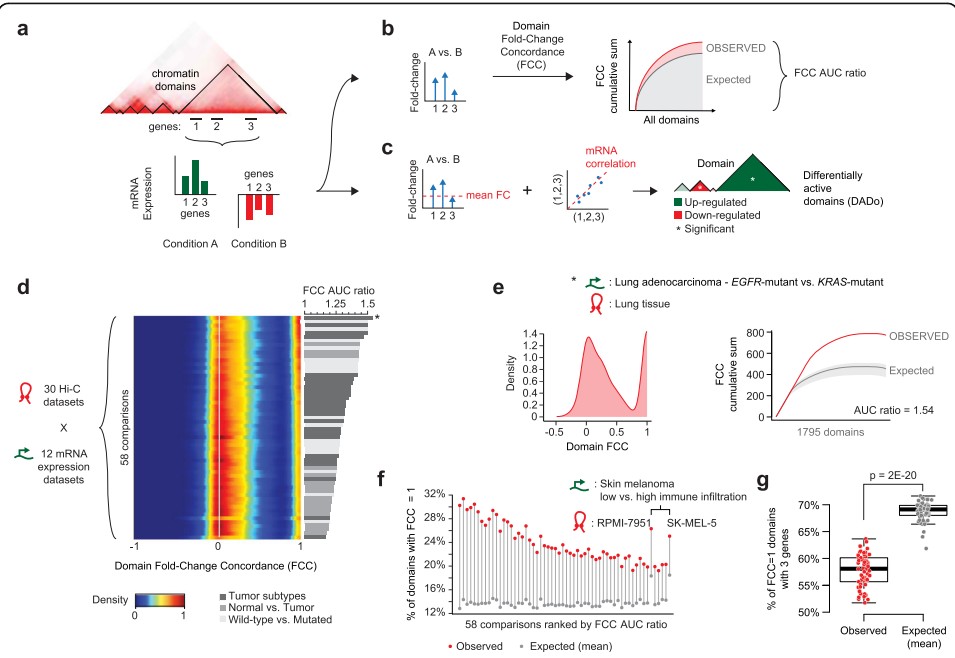

**Fig. 1** Gene co-regulation in chromatin domains. **a–c** Schematic of the DADo algorithm: **a** DADo integrates Hi-C (top: example of Hi-C contact map) and gene expression data (bottom: toy barplot showing concordant expression) to assess when genes (e.g., 1, 2, and 3) within a chromatin domain (black triangle in the Hi-C map) exhibit coordinated expression differences between 2 conditions (e.g., A and B). **b** First, the fold-change concordance (FCC) score is computed and the ratio between the observed (red) and expected (gray) areas under the curve (AUC) are computed. **c** Next, the mean gene expression fold-change (FC) and mean mRNA correlation among all genes in the domain are computed and used to determine differentially active domains. **d** Heatmap representation of the density distribution of FCC scores (range: −1 to 1; *X*-axis) for each of the 58 comparisons (*Y*-axis) made by matching 30 Hi-C and 12 mRNA expression datasets (left). Barplot of the FCC AUC ratios for all comparison (right). Asterisk (*) indicates the top ranking dataset (lung tissue—lung adenocarcinoma (LUAD)—*EGFR*-mutant vs. *KRAS*-mutant—see panel **e**). **e** Example of the FCC distribution (left) and the cumulative sum curves (right) for the dataset with highest FCC AUC ratio (lung tissue—LUAD—*EGFR*-mutant vs. *KRAS*-mutant). **f** For each dataset (*X*-axis), the percentage of chromatin domains that are fully concordant (i.e., FCC = 1) is shown for the real data (red dots) and the randomized data (gray dots). **g** Ratio of fully concordant domains comprising only 3 genes for the observed (red, left) and average of permutation data (gray, right).

FCC scores are equal to 1 in case of full concordance, i.e., all genes in *t* change in the same direction, are proximal to 0 when sign and magnitude of fold-changes exhibit no concordance patterns, and assume values close to −1 when most genes have concordant fold-changes, but the few that are discordant exhibit significantly higher absolute fold-changes than the others (Additional file 1: Fig. S1a). Once FCC scores are computed for all domains, they are ranked in descending order and the cumulative sum curve of ranked FCC scores is compared to the one obtained after permuting gene-to-domain assignments (Fig. 1b), with permutations occurring only within the same expression quintile. The ratio between the area under the curve (AUC) defined by the observed FCC values and the AUC defined by random FCC values can then be used to determine whether expression differences are more concordant within domains than expected (i.e., AUC ratio > 1, Fig. 1b).

To address the second question, we designed a statistical approach to determine dif-ferentially active domains (DADo). This approach integrates two tests to assess the extent of differential expression and correlation of gene expression within a domain

(Fig. 1c). Precisely, first, we compute the mean gene expression fold-change of genes within a given domain and derive an empirical $p$ value for these fold-changes based on gene-to-domain permutations. Next, we assess whether mRNA expression values for genes within a given domain exhibit greater correlation among them than with neighboring genes separated by a chromatin domain boundary (detailed procedures and null models are described in the "Methods" section). Empirical $p$ values obtained through the two tests are combined using the Stouffer's method and corrected for multiple testing using the Benjamini-Hochberg procedure. Finally, the DADo algorithm returns a list of significantly differentially active chromatin domains (adjusted $p$ value ≤ 0.01) between the two compared conditions. This approach can be related to popular differential mRNA expression analyses, where, instead of a gene, the chromatin domain is the unit of analysis.

### Concordant expression changes in chromatin domains

Given contrasting evidence supporting concordant gene expression and gene expression changes within chromatin domains, we reasoned that this concordance might be context dependent and not ubiquitously observed. To test this hypothesis, we decided to apply our approach across multiple datasets and compare different conditions for each dataset. We focused our analyses on cancer gene expression cohorts given the availability of large sample cohorts that were uniformly processed and analyzed and that exhibited high transcriptional heterogeneity across multiple conditions [23]. Specifically, we selected 12 tumor types with more than 90 samples from The Cancer Genome Atlas (TCGA) data cohort (https://www.cancer.gov/tcga) (Additional file 2: Table S1). To match each cohort with Hi-C data derived from the closest possible tissue or cell line, we collected and analyzed 30 Hi-C datasets. Given the distinction between compartment domains and TADs [13], here we used the Calder algorithm [10] to infer compartment domains, and TopDom to infer TADs [24]. Overall, domains identified by these tools were often coincident [10] and, as we will show, DADo analyses based on either compartment domains or TADs led to largely similar results and conclusions. Given the consistency observed among these tools, the results discussed in the following will refer to chromatin domains identified by Calder, unless explicitly indicated. When multiple Hi-C datasets were available for the same tumor type, we separately analyzed domains inferred from each Hi-C dataset, so to be able to test the robustness of our results. In total, we performed 58 analyses including comparisons between tumor subtypes (e.g., lung tumors from smokers vs. never-smoking patients), normal and tumor tissues (e.g., normal lung vs. lung tumor tissues), and tumors exhibiting or not a specific somatic mutation (e.g., lung tumors harboring a *KRAS* mutation vs. lung tumors wild-type for the *KRAS* gene) (Additional file 2: Table S1). In all comparisons, only chromatin domains comprising at least 3 genes were retained.

Across all comparisons, we invariably found AUC ratios greater than 1 (Fig. 1d—barplot), indicative of greater concordance of gene expression changes within domains than expected by chance. Comparisons between the same pairs of conditions but using domains inferred from different Hi-C datasets were always correlated both in terms of domain adjusted $p$ value (mean Pearson's correlation = 0.73, Additional file 1: Fig. S1b) and domain ranking (mean Pearson's correlation = 0.75, Additional file 1: Fig. S1c),

confirming the robustness of our results. Notably, in several cases, AUC ratio values were only moderately above 1. The overall distributions of FCC values were characterized by two major peaks: one around 0 (lack of concordance) with a larger tail towards positive values than what observed in the random distributions (Additional file 1: Fig. S1d) and a second peak of values close to 1 (high concordance) (Fig. 1d—heatmap). A representative example of this bimodal distribution was shown by the top ranking comparison, i.e., lung tumors driven by mutations of either the *EGFR* or the *KRAS* oncogenes, which characterize two distinct genomic subtypes of lung adenocarcinoma [25] (Fig. 1e). The discriminative factor between top and bottom ranking comparisons was the number of chromatin domains exhibiting completely concordant expression changes within their boundaries (FCC = 1, Additional file 1: Fig. S1e). Although these were typically a minority, the percentage of domains with FCC = 1 (termed *fully concordant domains*) was consistently higher than expected (Fig. 1f). Complete concordance of gene expression changes is more likely to occur when only a few genes are considered, e.g., gene pairs or triplets are expected to have concordant fold-changes 50% and 25% of the times, respectively. To test whether fully concordant domains included a larger number of genes than expected, we compared the percentage of fully concordant domains that were composed by 3 genes only (smallest possible size). This percentage was significantly higher in the randomized datasets than in the real ones (Fig. 1g). Overall, these results suggest that expression changes are more concordant within chromatin domains than expected, but this concordance is not a genome-wide phenomenon but rather restricted to a fraction of domains exhibiting high or even full concordance.

### Differentially active domains in cancer

The observed high concordance of gene expression changes within a limited yet greater than expected number of chromatin domains prompted us to statistically assess which domains are differentially active between two conditions. We applied the DADo algorithm to each of the 58 comparisons and identified between 2 and 61 differentially active domains in each comparison (adjusted *p* value ≤ 0.01—Additional File 3: Table S2 and Fig. 2a). To assess whether a similar number of domains would be obtained by simply selecting similarly sized genomic regions but crossing a domain boundary, we designed domain partitions for each comparison by generating "artificial" boundaries placed exactly in between two "real" boundaries (see the "Methods" section). On these artificial domains, DADo returned a lower number of significant hits (Fig. 2b and Additional file 1: Fig. S2a). These results further demonstrated that gene co-regulation within chromatin domains is more frequent than expected, even accounting for genomic distances among genes. We next assessed the relationship between differentially active domains detected by our approach and differentially expressed genes [26]. Interestingly, in each comparison, we found that a large fraction of significant domains did not comprise genes among the top 100 differentially expressed (Fig. 2a—red bars), suggesting that differentially active domains provide complementary information to standard gene differential expression analyses. In addition, whereas the percentage of differentially expressed genes (adjusted *p* value ≤ 0.01) could vary significantly among comparisons (from 0 to >75% of the analyzed genes), DADo returned an overall similar

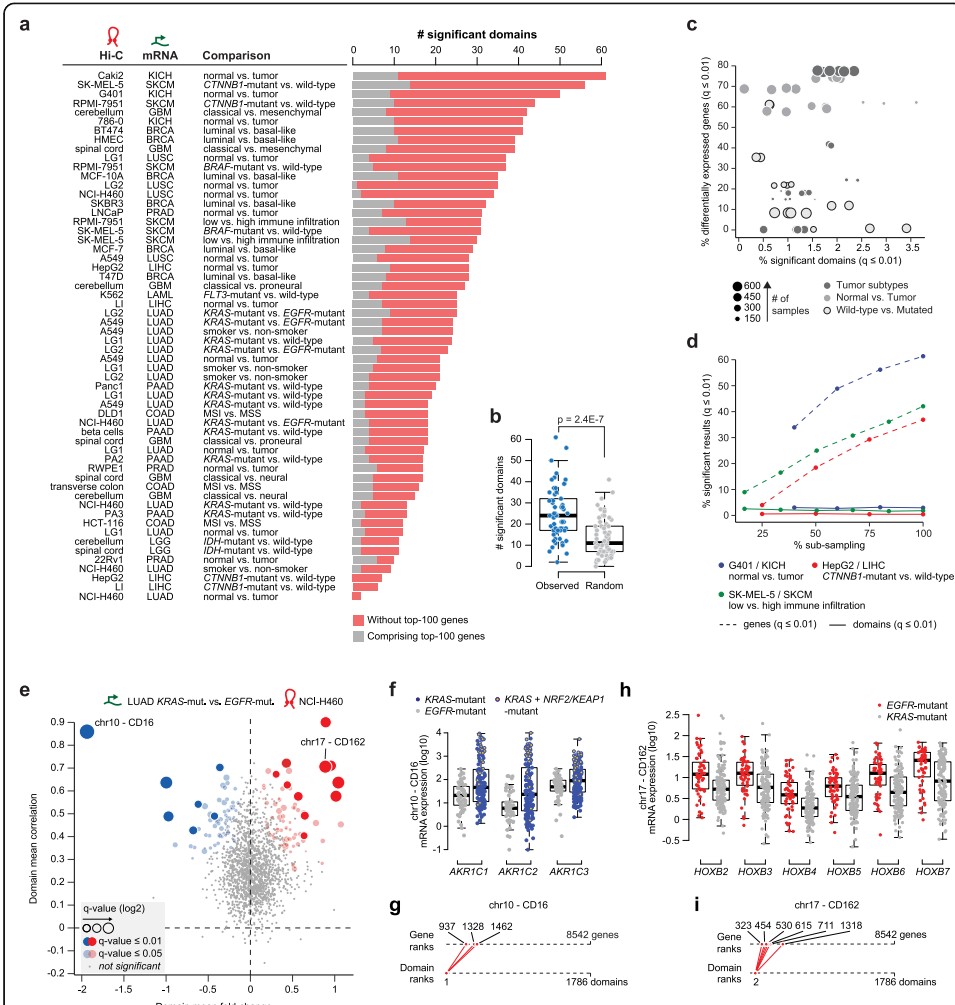

**Fig. 2** Differentially active domains. **a** Number of differentially active domains identified (X-axis) for all comparisons (Y-axis), ranked by decreasing number. For each comparison, we report the ID of the Hi-C dataset, mRNA expression dataset, and conditions being compared. The fraction of domains comprising any of the top 100 differentially expressed genes is shown in red (in gray otherwise). **b** Number of differentially active domains (Y-axis) in the observed (blue, left) and random data (gray, right). **c** Percentage of differentially expressed genes (Y-axis) and differentially active domains (X-axis) in each comparison. The size of the dot is proportional to the number of samples available for the analysis and the color of the dot indicates the category of comparison. **d** Percentage of differentially expressed genes (dashed-lines) and differentially active domains (solid lines) across different levels of sub-sampling (X-axis) for 3 selected datasets (color coded). **e** Domain mean gene expression fold-change (X-axis) and domain mean gene expression correlation (Y-axis) for the comparison between KRAS-mutant and EGFR-mutant lung adenocarcinoma (LUAD) (Hi-C data from the lung cancer cell line NCI-H460). Size of the dot is proportional to significance of the chromatin domain, blue (red) indicates negative (positive) average fold-change. Highlighted are two chromatin domains (CD) selected as a case study (chr10-CD16 and chr17-CD162). **f** Boxplot comparison of mRNA expression values (log10) for the genes belonging to chr10-CD16. KRAS mutant samples are color coded based on whether they also exhibit either NFE2L2 (NRF2) or KEAP1 mutations (yellow) or not (blue). EGFR mutant samples are in gray. **g** Top line represents the gene ranks based on differential expression analysis. Bottom line represents rank of the domain obtained from DADo. Genes from chr10-CD16 are shown in red. **h** Boxplot comparison of mRNA expression values (log10) for the genes belonging chr17-CD162. EGFR mutant samples are in red KRAS mutant samples are in gray. **i** Top line represents the gene ranks based on differential expression analysis. Bottom line represents rank of the domain obtained from DADo. Genes chr17-CD162 are shown in red.

number of significant hits (Fig. 2c) and was less affected by varying sample size than standard differential gene expression analysis (Fig. 2d).

To assess the dependency of our results on the domain identification approach or Hi-C data resolution (i.e., total number of reads), we re-analyzed all comparisons, either using the TopDom algorithm [24] to call TADs, or using in all comparisons the same set of domains identified in the GM12878 cell line, which was analyzed by Hi-C at the highest resolution [4]. FCC AUC ratios derived using Calder or TopDom were highly concordant (Additional file 1: Fig. S2b) and so were the chromatin domain significance ranks returned by DADo (Additional file 1: Fig. S2c). Similarly, we obtained highly concordant results using domains from the GM12878 cell line (Additional file 1: Fig. S2d,e), indicating that our results are robust to changing domain caller or Hi-C data resolution.

To examine in more detail the differentially active domains that we identified, we focused on the comparison between KRAS-driven and EGFR-driven lung adenocarcinoma, which exhibited the highest AUC ratio (Additional file 2: Table S1). Lung adenocarcinoma expression data were matched with Hi-C data generated from a normal lung tissue (LG1 and LG2) and the NCI-H460 and A549 lung cancer cell lines. Using NCI-H460 cells, we found 11 chromatin domains exhibiting concordant upregulation in EGFR-driven tumors (Fig. 2e, adj. *p* value ≤ 0.01—red dots) and 7 domains exhibiting concordant upregulation in KRAS-driven tumors (Fig. 2e, adj. *p* value ≤ 0.01—blue dots). The most significant chromatin domain (CD16) included 3 members of the Aldo-Keto Reductase (AKR) gene family (*AKR1C1, AKR1C2, AKR1C3*), located on chromosome 10p15 (Fig. 2f). AKR genes are oxidoreductases induced by the nuclear factor-erythroid 2-related factor 2 (NRF2, gene name: *NFE2L2*) and have been found consistently upregulated in the lung and other cancer types, especially in correspondence of mutations activating the NRF2 pathway [27]. NRF2 is over-activated in lung cancer either through mutations of the NRF2 encoding gene, *NFE2L2*, or loss-of-function mutations of *KEAP1*, whose protein product ubiquitinates and degrades NRF2 [25, 28, 29]. Mutations in either *KEAP1* or *NFE2L2* were found only in KRAS-driven tumor in our cohort and were associated with upregulation of all 3 AKR genes (Fig. 2f). However, even in the absence of *KEAP1* and *NFE2L2* mutations, KRAS-driven tumors exhibited concordant higher expression of all AKR genes within CD16 (Fig. 2f and Additional file 1: Fig. S3a). These results suggest a broad extent of NRF2 activation in KRAS-driven lung cancer, potentially associated with alternative mechanisms to known oncogenic mutations of the NRF2 pathway. Interestingly, AKR genes were not among the top differentially expressed genes between KRAS- and EGFR-driven lung tumors, with only *AKR1C1* barely passing the top-1000 cutoff (Fig. 2g). The second most significant domain (CD162) comprised 6 Homeobox B (HoxB) transcription factor gene family members, which were all upregulated in EGFR-driven tumors (Fig. 2h). HoxB genes were again not among the top differentially expressed genes (Fig. 2i). HoxB genes are transcription factors involved in development [30] and whose activation is regulated by chromatin domain boundaries [31, 32]. Recently, Hox genes have been shown to transcriptionally activate *EGFR* in drosophila [33] and breast cancer cells [34], although their role in EGFR-driven lung tumors is largely uncharacterized. Among other significant domains, we found several cancer-associated genes. These include the hepatocellular carcinoma-related protein 1 (HCRP1, gene name: *VPS37A*) and the

phosphoinositide (PIP) phosphatase *MTMR7*, which were downregulated in EGFR-mutated lung tumors, consistent with their ability to inhibit EGFR phosphorylation and signaling [35–38], the hypoxia-associated genes *NBN* and *OSGIN2* [39, 40], and the tumor suppressive gene cluster *TNSRSF10*(A/B/C/D), which were downregulated together with *RHOBTB2*, another tumor suppressor gene [41, 42]. These results showed that while standard differential expression analyses are designed to capture strong expression differences of individual genes treated as independent variables, differentially active domains can reveal moderate but concordant expression differences of co-regulated genes, potentially highlighting previously missed mechanisms of oncogene regulation.

Intriguingly, among the 18 domains that were differentially active between EGFR- and KRAS-mutated lung tumors, we found several domains comprising genes involved in immune pathways (Additional file 3: Table S2). These domains included two HLA class II gene clusters, CD1 dendritic cell marker genes (*CD1A*, *CD1C*, *CD1E*), interferon alpha and beta receptor subunits (*IFNAR1* and *IFNAR2*), and interferon-induced proteins (*IFIT1*, *IFIT2*, *IFIT3*). All these domains were found downregulated in KRAS-mutated lung tumors compared to EGFR-mutated cases. By retrieving *tumor purity scores* computed by integrating independent lines of evidence [43], we found that indeed TCGA samples derived from KRAS-mutated lung tumors had higher purity scores than EGFR-mutated tumors (Additional file 1: Fig. S3b). These results suggested that expression differences within these chromatin domains might here be driven by a different extent of immune infiltration.

Upon exploring chromatin domains that scored as significant across multiple comparisons, we found a subset of domains that was over-represented among significant results, with some domains appearing as significant in up to 27 (~50%) comparisons (Additional file 1: Fig. S4a). Gene set enrichment analysis revealed that genes in these domains were largely associated to immune pathways and immune cell markers (Fig. 3a and Additional file 4: Table S3). Next, we correlated expression of genes within each domain with purity scores retrieved for all TCGA samples and, for each domain, we computed the mean correlation. Strikingly, negative correlations were highly enriched among differentially active domains (Fig. 3b) indicating that gene expression in these domains was driven by immune infiltration rather than cancer cell intrinsic expression differences. We flagged chromatin domains with correlation in the lowest 5% of the overall distribution as activated in immune cells (*immune domains*). Interestingly, although immune domains accounted for less than 10% of all domains in each comparison, they frequently represented between 20 and 60% of significant domains (Fig. 3c). These results indicate that immune cell markers are frequently co-regulated genes more likely to be found within a same domain than expected. To test whether the overall significant co-regulation within chromatin domains that we previously observed (Fig. 1d—barplot) was exclusively due to immune domains, we re-analyzed all datasets with DADo, after having excluded immune domains from the analysis. As expected, the new AUC ratio values were lower than those originally computed (Fig. 3d), although all of them remained greater than 1. Similarly, the numbers of significant differentially active domains were now smaller in each comparison (Additional file 1: Fig. S4b) but remain significantly higher than those obtained with

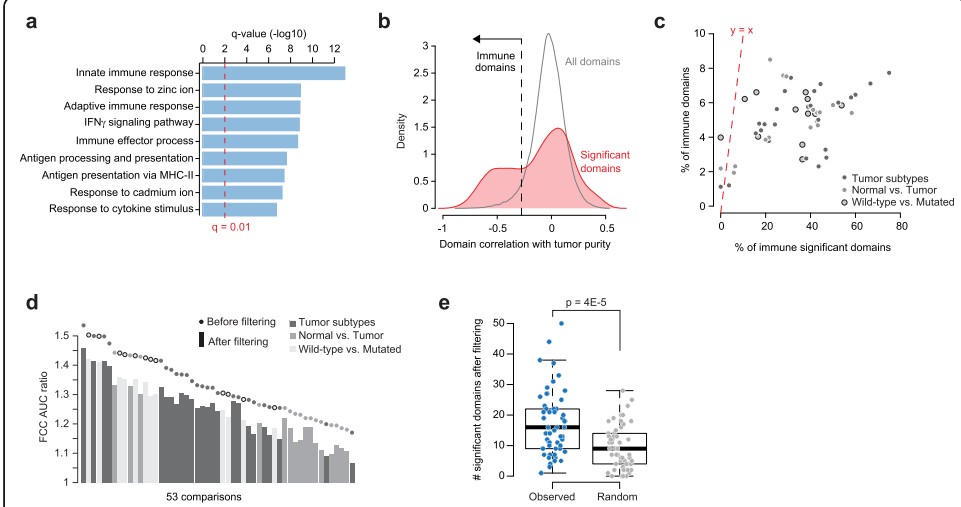

**Fig. 3** Immune chromatin domains. **a** Gene ontology (GO) categories significantly enriched for genes in domains that were found differentially active in multiple datasets (n > 7). **b** Distribution of the average correlation of gene expression with tumor purity for all domains (gray) and differentially active domains only (red). The dashed line indicates the 0.05-quantile of the distribution for all domains, used as a threshold for the detection of immune domains. **c** Percentage of immune domains among all domains (Y-axis) and among differentially active domains (X-axis). The dashed red line indicates the y=x line. Each dot is a dataset, the color indicates the comparison category. **d** FCC AUC ratios obtained after removing immune domains (barplot) versus FCC AUC ratio values obtained without filtering (dots). Color indicates the comparison category. **e** Comparison of the number of differentially active domains in the observed and random domain partitions after removing immune domains.

random partitions (Fig. 3e and Additional file 1: Fig. S4c). Overall, co-regulation of gene expression within chromatin domains is prominent among markers of immune cell types and analyses of co-regulation within chromatin domains in cancer are thus invariably affected by tumor purity. Nonetheless, independent of the extent of immune infiltration, differential activation of specific chromatin domains was more frequent than expected.

Beyond immune cell pathways, we observed that several significant domains comprised genes belonging to a same family, consistent with these genes being frequently co-regulated. Indeed, using gene family annotations for 941 gene families from the HUGO Gene Nomenclature Committee (HGNC - https://www.genenames.org/), in 34.4% of significant domains a unique gene family was represented, as opposed to 4.8% of non-significant domains (Fig. 4a). Concordantly, domains comprising genes from more than 2 gene families were less represented among significant domains than among non-significant ones (Fig. 4a). To test whether genes belonging to the same family were overall more likely to be found in the same domain, we generated a gene network for each gene family, where gene family members were connected if they were found in the same domain. The numbers of edges in these networks were then compared to the number of edges found in networks comprising the same number of genes, but that were randomly sampled while preserving the same extent of gene proximity along the DNA sequence (see the "Methods" section). Strikingly, comparisons for all gene families showed that on average gene family networks had up to one order of magnitude more edges than random gene networks (Fig. 4b). These results showed that genes from a same gene family are significantly more likely to be found in the same

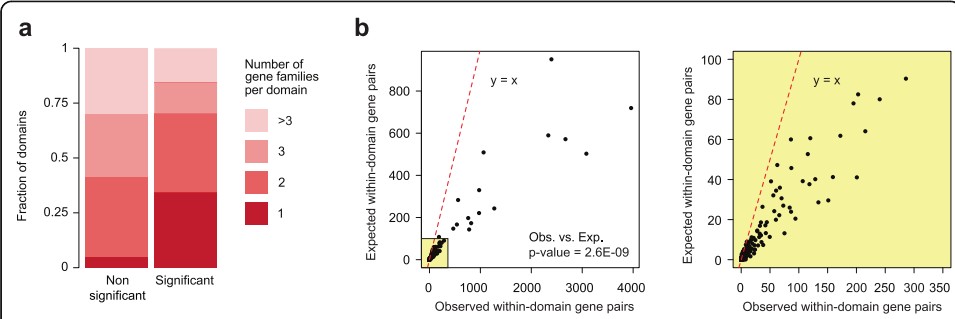

**Fig. 4** Differentially active domains are enriched for gene families. **a** Fraction of domains comprising genes annotated for a different number of gene families (1, 2, 3, >3) in non-significant (left) and significant (right) differentially active domains. **b** Expected (*Y*-axis) vs. observed (*X*-axis) number of edges in gene family networks (averaged by family: each dot corresponds to a family). Dashed red line indicates the *y=x* curve. Scatterplot with all values is on the left, yellow-square inset is zoomed in on the right

chromatin domain than genes in different families, suggestive of an evolutionary association between gene family formation and chromatin 3D structure.

## Chromatin structural features of differentially active domains

To assess whether differential activity within chromatin domain is associated with different long-range as well as local chromatin interactions, we decided to focus on a subset of comparisons between normal and tumor tissues (n = 7) for which tissue-matched Hi-C datasets were available for both conditions (Fig. 5a).

First, we compared domain boundaries in Hi-C datasets from normal and tumor cells and found that, on average, 79% of domain boundaries were shared among each pair of matched datasets, compared to an average of 75% among all pairwise comparisons in our collections of Hi-C datasets (n = 435). Even though the percentage of shared boundaries was lower for differentially active domains than other domains, 76.5% vs. 79% respectively (Fig. 5b), this decrease was modest and not significant based on random re-sampling of the same number of domains (*p* = 0.11, Fig. 5c). Next, we computed and compared chromatin compartments (A and B), sub-compartments (n = 8), and compartment domain ranks using the Calder algorithm [10]. Specifically, Calder assigns a "rank" varying between 0 (most inactive sub-compartment) and 1 (most active sub-compartment) to each domain and genomic bin, and the rank difference between matching bins or domains can be used to assess sub-compartment repositioning [10, 44]. Interestingly, we found that differentially active domains were 2-to-3 times more likely to change compartment (A to B or B to A) than other domains (Fig. 5d) and this was further highlighted by the distribution of compartment domain rank differences that, for differentially active domains, exhibited a long tail of high absolute rank differences (Fig. 5e), consistent with sub-compartment repositioning.

Upon extending the (sub-)compartment analysis to all Hi-C datasets, we found that differentially active domains identified by DADo across all comparisons were over-represented in the B compartment and sub-compartments (Fig. 5f-g). Indeed, although most domains were in the A compartment, consistent with its greater gene density, 23% of differentially active domains were in the B compartment compared to only 9% of all domains (Fig. 5f). This enrichment was largely due to a greater intra-domain gene

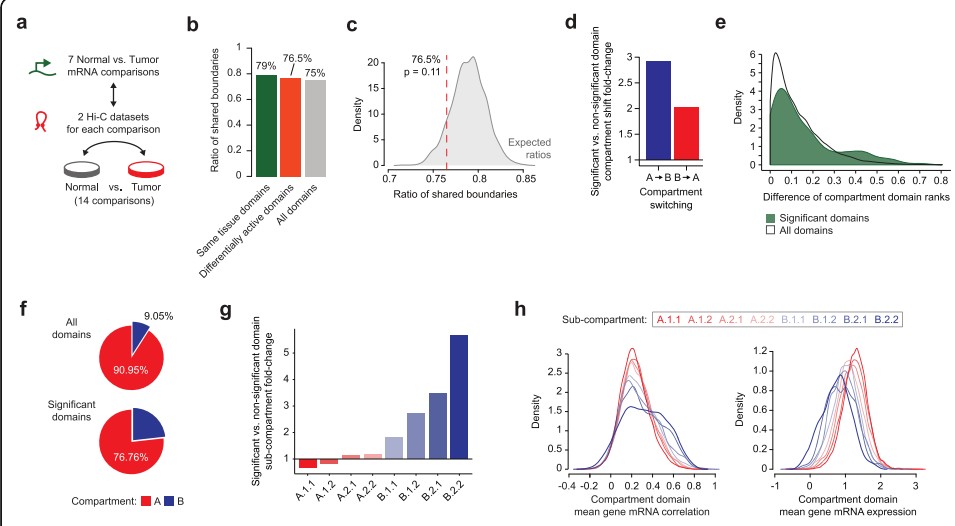

**Fig. 5** Chromatin structural features of differentially active domains. **a** Schematic of the comparisons: 7 mRNA expression comparisons between normal and tumor samples from the same tissues were matched with Hi-C datasets from normal and tumor cells of the corresponding tissue. DADo was run on chromatin domains determined from both Hi-C datasets for a total of 14 comparisons. **b** Ratio of shared boundaries between domains inferred from normal and tumor cells from the same tissue (green), only differentially active domains determined from these comparisons (orange), and all chromatin domains inferred from all Hi-C datasets that we analyzed (gray). **c** Distribution of the ratios of shared boundaries between domains inferred from normal and tumor cell Hi-C datasets. The distribution was built by 1000 random sampling of $n = 446$ domains. The observed ratio obtained for the 446 differentially active domains is shown by the red dashed line (76.5%) obtaining an empirical $p$ value = 0.11 from the expected distribution of ratios. **d** Fold-changes between the number of significant differentially active domains changing from A to B (blue) or B to A (red) compartment in the normal vs. tumor comparisons and non-significant domains, as determined by DADo. **e** Distribution of the domain rank difference of matching chromatin domains in normal and tumor Hi-C datasets for all chromatin domains (black line) and differentially active domains only (green density distribution). **f** Percentages of all chromatin domains (top pie chart) and significant differentially active domains (bottom pie chart) that are in the A (red) or B (blue) compartments. **g** Fold-changes between the number of significant differentially active domains and non-significant domains, as determined by DADo, that are in each of the 8 chromatin sub-compartments inferred by Calder. **h** Distributions of mean intra-domain gene expression correlations (left) and mean intra-domain gene expression levels (right) in each of the 8 chromatin sub-compartments inferred by Calder

expression correlation for genes in the B sub-compartments (Fig. 5h—left), which also exhibited lower mRNA expression on average (Fig. 5h—right). These results suggest that co-regulation of genes within the same chromatin domain is more common among lowly expressed or less efficiently transcribed genes. However, large datasets with matched Hi-C and mRNA data will be needed to systematically test this hypothesis.

Lastly, we explored intra-domain contacts within differentially active domains between two conditions, and whether these changes were concordant with cell transcriptional and epigenetic features. To this purpose, we selected the prostate cancer vs. normal prostate case study, where matching Hi-C, mRNA, and chromatin immunoprecipitation and sequencing (ChIP-seq) data were available for normal prostate (RWPE1) and prostate cancer (22Rv1) cell lines. Specifically, we analyzed ChIP-seq data for histone 3 lysine 27 acetylation (H3K27ac), an epigenetic marker associated with active enhancers and actively transcribed promoters. DADo identified in total 16 differentially active domains (adjusted $p$ value < 0.01) between normal and tumor prostate samples from TCGA (Fig. 6a). Among the 4 most significant domains, we found two domains on chromosome 17 that were less than 7 Mb apart and changed in opposite

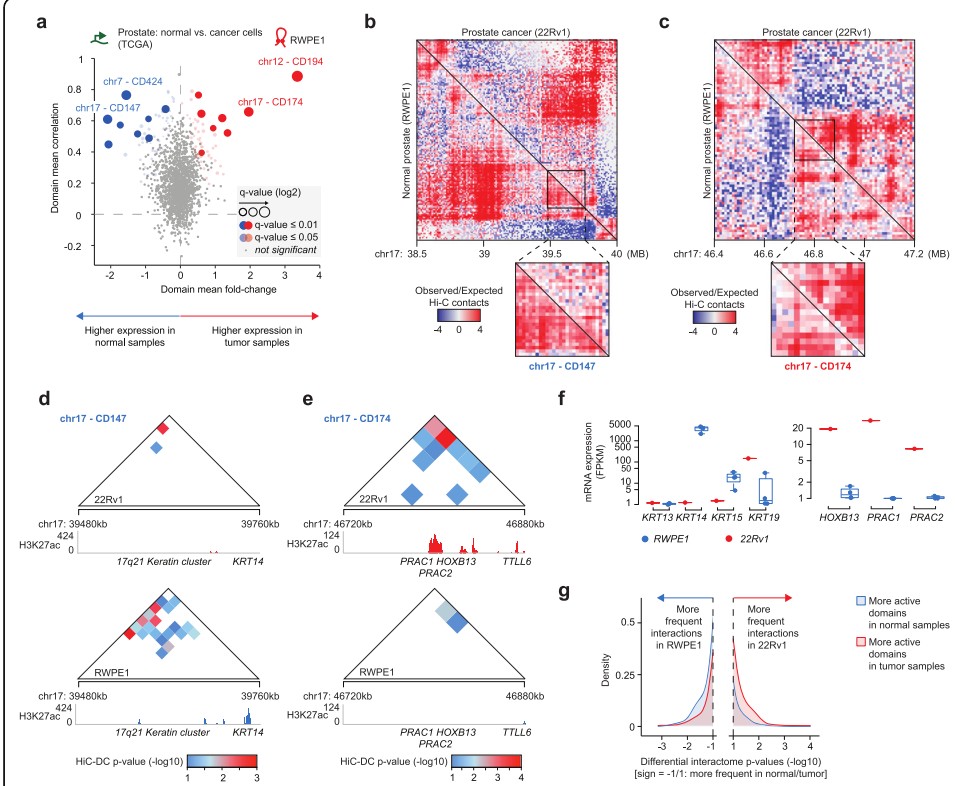

**Fig. 6** Differentially active domains in normal vs. tumor prostate samples and cell lines. **a** Domain mean gene expression fold-change (*X*-axis) and domain mean gene expression correlation (*Y*-axis) for the comparison between prostate cancer and normal prostate tissue samples (Hi-C data from the lung cancer cell line RWPE1). Size of the dot is proportional to significance of the chromatin domain, blue (red) indicates negative (positive) average fold-change of significant domains. Highlighted are the top 4 chromatin domains determined by DADo. **b**, **c** Observed vs. expected Hi-C contact maps for genomic regions in chromosome 17 comprising the differentially active domain CD147 (**b**) and CD174 (**c**). The lower triangular contact maps correspond to the RWPE1 cell line, and the upper triangular contact maps correspond to the 22Rv1 cell line. The contact maps corresponding to the two domains are zoomed at the bottom to improve visibility. **d**, **e** Significant interactions estimated with the HiCDC algorithm in CD147 (**d**) and CD174 (**e**) in 22Rv1 (top) and RWPE1 (bottom) cell lines. HiC-DC *p* values < 0.1 (−log10(p) = 1) are color coded, white cells correspond to HiCDC *p* values > 0.1. ChIP-seq tracks for H3K27ac are shown below each map. **f** Gene expression values of the genes in CD147 (left) and CD174 (right) in RWPE1 (blue, 4 replicates) and 22Rv1 (red, 1 replicate). **g** Distribution of *p* values (-log10(p)) obtained from differential interactome analyses of Hi-C contacts within differentially active domains, separately shown for domains that were found more active in normal samples (blue) and in tumor samples (red). The sign of the −log10(p) values was set to positive (negative) for interactions that were more frequent in the tumor (normal) Hi-C dataset

directions, hence providing the opportunity of assessing changes in intra-domain contacts in a relatively small region with similar sequencing coverage. CD147 (chr17: 39,480,000-39,760,000) was more active in normal samples, whereas CD174 (chr17: 46,720,000-46,880,000) was more active in tumor samples. By comparing the observed vs. expected contact frequencies in these regions, we found that CD147 and CD174 exhibited different contact frequencies in RWPE1 and 22Rv1 cell lines, whereas neighboring regions were remarkably similar (Fig. 6b-c). In particular, we observed that intra-domain contact frequencies were greater in the condition where the domain was more active. To quantify this observation, we computed significant chromatin interactions using the HiC-DC algorithm [45] and verified that CD147 exhibited more significant interactions in RWPE1 (Fig. 6d—bottom) than in 22Rv1 cells (Fig. 6d—top), whereas

CD174 had more significant interactions in 22Rv1 (Fig. 6e—top) than in RWPE1 (Fig. 6e—bottom). Importantly, a higher number of significant interactions was consistent with higher activity in the domain, as inferred by H3K27ac peaks (Fig. 6d–e, tracks below triangular maps) and matching mRNA expression data in the cell lines (Fig. 6f). To confirm these results beyond the CD147 and CD174 domains, we performed a differential interactome analysis [20, 44] that detected significantly different contact frequencies between the two prostate cell lines. Nicely, even though differential activity was estimated from the comparison of TCGA human samples (Fig. 6a), we found that domains that DADo found as more active in normal prostate samples also comprised a greater fraction of significantly more frequent interactions in normal prostate cells (RWPE1, Fig. 6g—blue curve). Conversely, domains that were more active in prostate cancer samples comprised a greater fraction of significantly more frequent interactions in prostate cancer cells (22Rv1, Fig. 6g—red curve). Overall, these results showed that differentially active domains reflect different local intra-domain interactions, which increase in frequency with increased H3K27ac and mRNA expression levels.

## Discussion

The possibility of unbiasedly exploring chromatin spatial interactions has provided a new perspective to understand and investigate gene regulation in normal and malignant cells. In cancer, for example, upregulation and downregulation of oncogenes and tumors suppressors are respectively frequently observed. By exploring the chromatin 3D structure of cancer cell lines, we and others have shown that oncogenic transcriptional changes can be attributed to disrupted chromatin domain boundaries [17, 46] or chromatin domain inactivation [20], the latter driven by repressive histone marks altering regulatory interactions between gene promoters. However, the relationship between chromatin structure and transcriptional regulation remains unclear.

Here, through an unbiased analysis of transcriptional changes across multiple tissues and tumor contexts, we showed that these are significantly associated with the chromatin structure, although this association is evident only in a small subset of chromatin domains, which varied among comparisons. What are the determinants of coordinate regulation within a domain remains an open question. The moderate transcriptional changes induced by genome wide loss of structural loops upon CTCF or cohesin depletion indicate that this loss is not sufficient to broadly rewire regulatory interactions and additional mechanisms are required [5, 6]. These likely include an active chromatin state (i.e., presence of active histone marks) and transcription factor binding to enhancer and/or promoter regions. Histone post-translational modifications are associated with chromatin compartmentalization and formation of (sub-)compartment domains; hence genome wide changes of histone marks might alter structure and transcription in a more substantial manner than altered CTCF and cohesin binding [44]. It will be interesting in the future to systematically explore the association between differentially active domains and matching histone mark profiles and use targeted approaches such as promoter-capture Hi-C or HiChIP to probe in more detail regulatory interactions in these domains.

This type of analyses is of particular interest in cancer, where histone modifications are frequently altered as a consequence of somatic mutations and/or cell plasticity. In a tumor, however, transcriptional signals often come from heterogeneous sub-populations of both tumor and non-tumor cells [43]. In our analyses, we found that infiltration of immune cells was a major determinant of differential activity in chromatin domains. Indeed, several immune cell markers formed co-regulated gene clusters often found within the same chromatin domain. Beyond immune infiltration, gene co-regulation and chromatin domain identification are likely to be affected by the intrinsic heterogeneity of tumors from different patients and tumor cells within the same tumor. A limitation of our study is indeed the lack of multiple instances where mRNA and Hi-C experiments were performed on the same sample. In particular, Hi-C analysis of multiple tumors of a given type would shed light on inter-patient heterogeneity of chromatin structures and how does this relate to the underlying genomic features of the disease. In addition, single-cell transcriptional and structural data will be needed to overcome the limitations posed by intra-tumor heterogeneity and understand to what extent transcriptional diversity is accompanied by structural diversity.

Lastly, our results showed an unexpected association between differentially active domains and B sub-compartments, which typically comprise genes with lower expression levels. This association was driven by an overall higher intra-domain gene correlation in these sub-compartments compared to A sub-compartments (Fig. 5h). These results suggest that gene co-regulation within chromatin domains is more relevant among less efficiently transcribed genes, whereas genes that are highly transcribed through strong activation of specific transcription factors are more likely to be transcribed independently of chromatin domain organization. This was especially evident for gene family members, where evidence of gene co-regulation in our data was particularly strong. Gene families formed during evolution by gene duplication events and often comprise gene clusters regulated through the same promoter [47]. Moreover, recent findings showed that paralogues from same TADs display higher correlation in gene expression patterns than those located in different domains [48]. Interestingly, our results suggest that clusters of genes from a same family are frequently within the same domain, more than expected by genomic proximity alone. Whether this is evidence of evolutionary constraints imposed by chromatin structural properties or, vice versa, it is gene evolution that determined chromatin conformation will need to be investigated.

## Conclusion

Our analysis supports concordance between gene co-regulation and chromatin domains in the context of cancer, driven by a subset of (almost) fully concordant domains. We proposed a new algorithmic approach, DADo, to systematically identify differentially active domains between two conditions. Overall, the identification of differentially active domains from the analysis of Hi-C datasets in multiple biological contexts can provide complementary information to standard differential expression analysis and ultimately inform on how coordinated regulation of specific gene sets determines cell phenotypes.

## Methods

### RNA-seq datasets

In this study, we used RNA-seq gene expression data generated by the TCGA Research Network (https://www.cancer.gov/tcga). In total, we used 20 RNA-seq datasets corresponding to a pair of conditions (Additional file 2: Table S1): 5 "normal vs. tumor" (e.g., kidney tumor vs. healthy kidney tissue), 7 "wild-type (wt) vs. mutant (mut)" (e.g., liver hepatocellular carcinoma with vs. without *CTNNB1* mutation) and 8 "subtypes" (e.g., luminal vs. basal breast cancer). For the differential expression analyses, "normalized results" (values divided by the 75-percentile and multiplied by 1000) of gene expression were used. These data were also used, after quantile normalization, for computing pairwise correlations. For building classes of expression for the gene-to-domain permutations as well as for plotting purposes, transcripts per million (TPM; "scaled estimates" * $10^6$) data were used. Only genes that had at least 80% of the samples with at least 5 reads were retained for downstream analyses.

### Specific sample annotation

Assignment of smoker and non-smoker patient status in lung adenocarcinoma was based on the clinical data annotation available on the TCGA website (https://portal.gdc.cancer.gov) and previously published data. To retrieve low and high infiltration samples of skin cutaneous melanoma (SKCM), we performed a gene set variation analysis using the *gsva* function from the *GSVA* R package [49]. For this purpose, we used the T cell infiltration signature, retrieving corresponding published lists of genes [50, 51]. Samples with a signature score lower than the first quartile of all scores were considered as lowly infiltrated, and those with a score higher than the third quartile as highly infiltrated. Additional tumor subtype annotations were retrieved from the corresponding TCGA publications.

### Gene-level differential expression analysis and domain-level averaged LogFC

Differential expression analysis was conducted on R with the *limma* package [26] (*lmFit* and *eBayes* functions). We retrieve the log2-fold-changes (logFC) and adjusted *p* values of the genes from the table returned by the *topTable* function.

### Generating Hi-C contact matrices

Hi-C intra-chromosomal contact matrices were either generated from raw fastq files or dumped from processed hic files using Juicer [52]. The Knight-Ruiz (KR) method was used for contact matrix normalization. In cases when KR normalization failed to converge, the VR (vanilla coverage) normalization was applied. For all Hi-C datasets, we generated contact matrices at 40 kb resolution. To eliminate technical noise, paired rows and columns that have more than 99% contact values being empty were removed for downstream analysis.

### Compartment domain calling

Chromatin compartments, sub-compartments, and compartment domains have been inferred using Calder [10]. In brief, Calder detects domains characterized by high intra-domain correlations of chromosome-wide intra-chromosomal interactions. Domains

identified by Calder are then hierarchically clustered based on inter-domain interactions and independently of linear proximity (i.e., proximity along the genome sequence). Chromatin sub-compartments are then determined from this domain hierarchy and can be analyzed at multiple levels of granularity. In principle, each internal node of the hierarchy can be thought as chromatin sub-compartment comprising all domains descending from it. Hierarchy branches are internally re-ordered based on gene density and without disrupting the clustering structure to match sub-compartments among different chromosomes. In this study, we considered either the top layer of the hierarchy, corresponding to the subdivision into A and B compartment, and the third layer, which determines 8 sub-compartments (4 within the A compartment: A.1.1, A.1.2, A.2.1, A.2.2 and 4 within the B compartment: B.1.1, B.1.2, B.2.1, B.2.2). Finally, a normalized rank value is assigned to each domain as its rank within the hierarchy sorted by increasing gene density normalized within the [0,1] interval. A detailed description of the method is provided in [10].

### Hi-C significant interactions

We used HiC-DC [45] to compute the statistical significance of chromatin interactions at bin level (bin size = 20 kb) for RWPE1 and 22Rv1 cell lines. For HiC-DC parameters, the degree of freedom in the hurdle negative binomial regression model was set as 6. We determined the sample size parameter by trying 20 values in the [0.5,1] range with equal distance and choosing the maximum value that did not resulted in optimization failure in R. Other parameters of HiC-DC were set as default. We observed that interaction $p$ values resulting from HiC-DC were systematically lower for Hi-C datasets with higher number of overall contacts. To correct for this bias, we conducted chromosome-wise sub-sampling of contacts, such that each chromosome had the same number of contacts for RWPE1 and 22RV1.

### Differential interactome analysis

Differential interactome analysis of a chromatin region of interest searches for interactions (pixels in its contact map) with a significantly different contact frequency between two conditions. For each pixel (bin1, bin2), we defined an interaction strength S as the -log10-transformed HiC-DC $p$ value. We then computed the difference of interaction strength between two conditions as $\Delta S = S1 - S2$ and tested $\Delta S$ for significant deviation from 0 with respect to a background distribution. We generated the background distribution by computing $\Delta S$ for all pixels within a 2 Mb window across all chromosomes. An empirical two-tailed $p$ value was obtained from this background distribution as $p = P(|\Delta S| > |\Delta S_{background}|)$. In each comparison, we also kept track of the "direction" of the significant difference ($-1$ or $1$), i.e., whether a given pixel was found more significantly frequent in the condition 1 or condition 2.

### Shared boundaries analysis

To determine the percentage of shared boundaries between two datasets, we tested if for a boundary located in bin X in one dataset, we could find a corresponding boundary in the other dataset within the bin interval [X-2, X+2] (i.e., tolerance radius = 2 bins or 80 kb). Given each boundary can determine either the start or end position of a

domain, we separately computed the shared percentages of start and end boundaries. The final percentage of shared boundaries was considered as the mean of the two percentages.

### The DADo algorithm

#### Data pre-processing

Our algorithm takes as inputs a list of chromatin domains (CDs), which can be either compartment domains or TADs pre-computed with a separate tool, one matrix of gene expression (genes x samples table), and sample annotations defining two conditions to be compared. First, genes are assigned to CDs based on their transcription start sites (TSS). If certain genomic regions are not covered by any CD and the TSS of a gene is not located within a CD, but its end (3'UTR) is inside a CD, then the gene is assigned to the CD that contain its end position. Gene coordinates are retrieved from the GFF file for the GRCh37 human genome build downloaded from Ensembl. As an additional filter, only domains containing at least 3 genes but not more genes than the 99th (dataset-specific) percentile are retained for further analyses.

**Step 1: FCC AUC ratio analysis** The first step of the algorithm aims at quantifying concordance of gene expression within chromatin domains across the entire genome. To this end, we first calculate a domain-level metric, the fold-change concordance (FCC) score, defined as in Eq. 1. The FCC score ranges between –1 (full discordance) and 1 (full concordance). Next, to derive a genome-wide quantification of this concordance (or lack thereof), domains are ranked by decreasing FCC and the cumulative sum is calculated. A similar procedure is adopted for the random data, for each of the 100,000 permutations (see next section) and the 95th percentile value is retained in order to derive a unique random cumulative sum curve. Finally, we calculate the ratio between the areas under the curve (AUC) of the observed and the random cumulative sum curves. The AUC is computed with the *auc* function of the *flux* R package (https://rdrr.io/cran/flux/).

**Step 2: Differentially active domains** Once the genome-wide concordance has been assessed, the next goal is to identify differentially active domains between 2 conditions. DADo assesses gene co-regulation in a given domain by computing 2 scores for each domain:

1) The mean gene expression fold-change (mFC) between the 2 conditions computed over all genes within the domain;
2) The mean gene expression correlation (mCor) in the analyzed dataset computed over all genes within the domain.

To compute mFC scores, DADo performs gene-level differential expression analysis with the *limma* R package (*lmFit* and *eBayes* functions) and retrieves log2-fold-changes (logFC) as well as adjusted *p* values of the genes (*topTable* function). Finally, for a given domain d, the logFCs of the genes in d are averaged to obtain mFC.

To compute mCor scores, for two given genes within a same domain, the Pearson's correlation of expression across all samples is used as a measure of co-expression. Correlations among all pairs of genes in a given domain are then averaged to obtain mCor.

For each metric, we separately evaluate statistical significance of each score by means of empirical null models (see next section) hence obtaining, for each domain, two empirical $p$ values. These $p$ values are combined into a single $p$ value. Combined $p$ values are adjusted for multiple testing and a domain is called "differentially active" if its adjusted combined $p$ value is lower or equal to 0.01.

***Gene expression fold-change statistical significance*** To assess the significance of the mFC score, we re-compute this measure after 100,000 gene-to-domain permutations generated following a procedure previously described [53]. Gene-to-domain assignments were permuted within 5 equal-sized classes generated based on the expression level. To account for gene length, scaled estimates of gene expression data (instead of RSEM) were used to create the gene expression classes. An empirical $p$ value was derived as the number of permutations returning an absolute mFC value greater or equal to the one observed divided by the total number of permutations (we add 1 to the numerator and denominator to avoid $p$ value = 0).

***Gene expression correlation statistical significance*** To assess the statistical significance of the mCor score, we re-computed this measure on randomized domain partitions generated according to the following strategy:

- For a domain D comprising k genes, we selected k genes from the left-side and right-side adjacent domains, starting from the closest ones to the domain boundaries;
- Next, we computed the average of the pairwise Pearson's correlations between each gene in the domain D and each of the genes that were sampled from the adjacent domains to obtain cross-boundary mCor values;
- Cross-boundary mCor values were computed for each domain and pooled for all comparisons that we performed to build a unique distribution from ~100,000 mean correlations;
- The empirical $p$ value for a given domain mCor score was then computed as the total number of cross-boundary mCor values greater or equal to the observed mCor score divided by the total number of cross-boundary mCor scores (we add 1 to the numerator and denominator to avoid $p$ value = 0).

The mFC and mCor empirical $p$ values were combined using the Stouffer's method (one-sided), and adjusted for multiple testing following the Benjamini-Hochberg's method (as implemented by the *p.adjust* R function). A chromatin domain was considered as differentially active (DA) if its combined $p$ value ≤ 0.01.

### Robustness of the analyses

To assess the robustness of our approach, we first repeated our analyses on the set of TADs resulting from the TopDom TAD calling method implemented in an R package

released at https://github.com/HenrikBengtsson/TopDom, with the parameter *window.-size=5*. In addition, all comparisons were repeated using a unique set of domains that were derived with Calder from the GM12878 Hi-C dataset, which has the highest data resolution (4.9 billion of reads).

### Effect of sample size on gene-level and domain-level analysis

To assess the effect of sample size on the detection of differentially expressed genes and differentially active domains, we selected one dataset from each comparison type and subsampled a certain number of samples (nSamp) while preserving the nSamp ratio between the two conditions (nSamp condition 1/nSamp condition 2/subsampling ratio): *G401 - KICH normal vs. tumor* (25/65/1, 20/52/0.8, 15/39/0.6, 10/26/0.4), *SK-MEL-5 - SKCM low inf. vs. high inf.* (119/119/1, 100/100/0.84, 80/80/0.67, 60/60/0.50, 40/40/0.34, 20/20/0.17) *HepG2 - LIHC CTNNB1wt vs. CTNNB1mut* (256/92/1, 192/69/0.75, 128/46/0.5, 64/23/0.25).

### Enrichment analysis

Gene ontology enrichment analysis was performed using the *enricher* function from the clusterProfiler R package [54] (with minGSSize=1 and maxGSSize=500). Gene sets of biological processes were retrieved from the Molecular Signatures Database (MsigDB; www.gsea-msigdb.org/gsea/msigdb; version 6.1.).

### Conserved region analysis

Significantly differentially active domains were extracted from all datasets (adj. $p$ value ≤ 0.01) and subjected to an all-versus-all matching. Only the matches with ≥ 80% base-pair overlap were retained. Duplicated and nested sets of matching domains were discarded. Conserved regions (starting at the smallest start and ending at the largest end of the matching domains) with at least 3 genes at the intersection of the matching domains were then retained. Finally, those with ≥ 80% of the intersect genes in common were merged. The GenomicRanges R package [55] was used for this analysis (for the genomic range matching). For the permuted data, we sampled for each dataset as many not differentially active CDs (adj. $p$ val. > 0.01) as differentially active CDs (adj. $p$ val. ≤ 0.01; 1000 permutations). Then, conservation analysis was conducted in the same way as for the observed data. For the gene ontology enrichment analysis of the conserved regions, we retain those that were conserved in at least 8 datasets.

### Association with tumor purity

Sample purity was retrieved from ref. [43]. For each gene, we computed the Pearson's correlation coefficient between gene expression (log10 TPM) and purity across all samples. These values were then averaged at the domain level. We defined as "immune domains" those domains with an average purity correlation below or equal to the 0.05-quantile of the non-significantly differentially active domains (here $r = -0.27$).

### Gene family data and family networks

Gene family data were downloaded from https://www.genenames.org/cgi-bin/genefamilies/download-all/tsv (April 2018). If a gene was annotated with multiple families, the first-level (more general) was retained (e.g., "Collagens" for "Collagens|Collagen proteoglycans"). To test whether genes from the same family were more likely to be within the same domain than expected, for Hi-C each dataset, we built a network for each gene family and connected genes if they were in the same domain (observed number edges). The expected number genes in the same domain (expected number of edges) were obtained by applying the same procedure after randomly sampling the same number of genes (among those having a family annotation). Genes were sampled by preserving the chromosome representation and by preserving gene contiguity: if k genes from a given family were clustered at contiguous genomic positions, then the same number of contiguous genes were randomly sampled (for each gene cluster). The sampling of the cluster was repeated if 80% or more of the sampled genes belong to the same family. Values were then aggregated by family and averaged over 100 permutations and across datasets and only domains where at least 50% of their genes were annotated for a given gene family were retained.

### Cross-boundary domain partitions

To assess the relevance of domain boundaries in determining gene co-regulation within a domain and differentially active domains, we generated "artificial" domain partitions such that each artificial domain has start and end positions corresponding to the mid-positions of the "real" CDs. In this way, each artificial domain traverses a real domain boundary and comprise genomic regions from two adjacent real domains. We computed these artificial partitions for all datasets and ran DADo on all comparisons to compare the number of significant hits that we obtain with the real and artificial domains. Since our analyses depend on the list of genes within a domain (rather than its coordinates), we exclude from this analysis artificial domains where all genes come from a unique real domain.

### ChIP-seq data

We downloaded from ENCODE (https://www.encodeproject.org) the fold change over control bigWig files (replicates 1,2) for the 22Rv1 (ENCFF282RLY) and the RWPE1 (ENCFF039XYU) cell lines. These files were then converted to bedGraph format with the bigWigToBedGraph tool (downloaded from http://hgdownload.soe.ucsc.edu/admin/exe/linux.x86_64/).

## Supplementary Information

---

**Additional file 1.** Contains Supplementary Figures S1-S4.

**Additional file 2.** Table S1. This table provides for each dataset: the number of samples, the FCC AUC ratio, the total number of chromatin domains, the number of differentially active (DA) domains (p<= 0.01), the total number of immune domains and the number of immune domains among the DA domains.

**Additional file 3.** Table S2. Complete set of results obtained by DADo on 58 comparisons.

**Additional file 4.** Table S3. Result of the gene set enrichment analysis for the domains differentially active in multiple comparisons.

**Additional file 5.** Review history.

---

### Acknowledgements
The authors wish to thank Elisa Oricchio for insightful discussions and the feedback provided during this project.

### Review history
The review history is available as Additional file 5.

### Peer review information

### Authors' contributions
M.Z. and G.C. conceived the study. M.Z. designed and performed all the analyses. Y.L., D.T., M.M. contributed to data analysis. G.C. wrote the manuscript with input from all the authors. All authors read and approved the final manuscript.

### Authors' information
Twitter handle: @CirielloLab (Giovanni Ciriello).

### Funding
This project was supported by Fondation Gabriella Giorgi-Cavaglieri and the Swiss League Against Cancer Foundation (KFS3983-08-2016).

### Availability of data and materials
Publicly available Hi-C datasets were retrieved from the Gene Expression Omnibus (GEO) repository (GSM1631184, GSE66733, GSE63525, GSE118514, GSM2809542, GSE109229, GSE118588, GSE99051, GSE105194, GSE105318, GSE105381, GSE87112) and the ENCODE repository (ENCSR079VIJ, ENCSR312KHQ, ENCSR346DCU, ENCSR401TBQ, ENCSR444WCZ, ENCSR489OCU, ENCSR549MGQ, ENCSR862OGI, ENCSR440CTR, ENCSR504OTV). RNA-seq datasets were downloaded from The Cancer Genome Atlas website (https://www.cancer.gov/tcga). H3K27ac ChIP-seq datasets used in this manuscript have been downloaded from ENCODE (ENCFF282RLY for 22Rv1; ENCFF039XYU for RWPE1). mRNA expression data for the cell lines have been downloaded from the Sequence Read Archive (SRA) (SRR5120472 for 22Rv1; SRR5558489, SRR5558490, SRR5558491, SRR5558492, SRR5558493 for RWPE1). The scripts used for our analyses are deposited in a GitHub [56] and Zenodo [57] repositories, released under the "Free Public License 1.0.0" open-source license.

## Declarations

### Ethics approval and consent to participate
Not applicable.

### Consent for publication
Not applicable.

### Competing interests
The authors declare that they have no competing interests.

### Author details
[1]Department of Computational Biology, University of Lausanne (UNIL), Lausanne, Switzerland. [2]Swiss Cancer Center Leman, Lausanne, Switzerland. [3]Swiss Institute of Bioinformatics, Lausanne, Switzerland.

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

## 
