## [**Additional file 5.** Review history. · Genome Biology]

Review History

First round of review

Reviewer 1

Are you able to assess all statistics in the manuscript, including the appropriateness of statistical tests used? Yes: Everything seems to have an appropriate background/null hypothesis and appropriate tests.

Comments to author:

The authors study an interesting but subtle point relating the correspondence between contact domains in genome architecture and the co-regulation of transcription, and more generally the possible role of self-interacting chromatin domains. Here the use of a large number of Hi-C datasets is welcome.

The analyses that are presented seem thorough enough, even if the results are not stand-out in terms of impact. However, overall the analysis seems fairly limited in its scope and ambition, and I have a few comments regarding clarity. Some effort has gone into identifying differentially active domains, but then I would have expected to see further follow-up regarding chromatin marks, genome accessibility, chromatin compartments etc., to get a better handle on the biological events that may be happening in the most significantly co-regulated/transcribed domains.

Also, the Hi-C data is only analyzed in terms of domain boundaries. I would be interesting to see if there are any interesting changes in the degree of intra- and inter-domain interaction probabilities (for example).

Given the high-degree of sequential (and hence intra-domain) co-localisation in homologous gene families it is important to dissect how much homology drives the occurrence of coordinated domain regulation. Similarly, the non-immune domains discussed in detail should probably not all be from homologous clusters, as they are currently (AKR and HOX families), to give more balanced examples.

Related to this, and in general, it would be better to see more exploration in the text of the biological relevance of differently active domains in multiple cell type comparisons (preferably relating to cancer specific changes rather than immunity). - In essence it would be good to have an overview of the lists presented in SuppTable 2 in terms of salient biological features, For example, it would have been helpful to see analyses relating to stratification of genes according to TF types, constitutive vs cell-type specific etc.

Overall it was fundamentally a good thing that immune infiltration was identified, but I feel any message becomes somewhat occluded if the cancer genes and how they relate to the differential activity, are not explored more fully.

Taking some biologically relevant examples, I would like to have seen some actual Hi-C maps in the most differential regions.

An idea: the authors may like to consider that 3D genome folding may have more relevance/assistance for the less efficiently transcribed, differently regulated genes via an optimization of enhancer and TF dynamics.

Some more specific points:

Around line 120. The nomenclature in the equation seems an odd; an somewhat complex mixture of textual labels and mathematical notation, which to me is not clear at first glance. For example I feel " $\#log_2 FC(t)$ " is an obscure way of representing the number of genes in the region. Also, I feel "log" should only be used as a clear mathematical function.

Also, the textual description that accompanies the FFC formulation is unclear. What is being summed over? Regarding "Negative log_2 " values; does this mean the selection of negative only values rather than multiplication by -1?

Line 126: What is the size of the quantiles (decile, percentile?)

Line 173. I think it is a bit of a stretch to say that the values are "highly" correlated.

Figure 1f & 1g. It is not clear in the main manuscript how the random expectation in 1g calculated and why this is so big compared to the somewhat lower values in 1f.

Fig2e-h The assessment of robustness is good, but could be supplemental to make room for further, more biological analyses.

Reviewer 2

Are you able to assess all statistics in the manuscript, including the appropriateness of statistical tests used? Yes

Comments to author:

One of the controversial questions in the field is whether the 3-D genomic structure regulates gene transcription. However, a systematic, comprehensive analysis of the correlation between 3D genome structures like TADs and sub-TADs, typically but not exclusively regulated by CTCF, and gene expression levels in multiple cell lines has not been performed. Zufferey and colleagues analyzed 30 different Hi-C experiments among 12 cancer cell lines to show that gene co-regulation with chromatin domains does exist albeit within a small number of domains. With further gene ontology analysis and tumor purity score analysis, the authors showed these active co-regulated domains are most strongly associated with immune response genes, suggesting these gene expression changes under the co-regulated chromatin domains is likely driven by immune infiltration. However, other interesting examples emerged. The contribution of this research is important as it provides evidence that gene expression correlates with chromatin structures. Indeed, it is our opinion that such co-regulation might be broader if finer structures were examined, as this has been reported in murine ES cells and elsewhere for specific examples.

The data in this work are rigorous. Zufferey and colleagues first introduce the methodology of "DADo" algorithm then perform it on datasets of different cell lines from TCGA to calculate the co-regulating gene sets within the same chromatin domain (Fig. 1). The number of such co-regulated domains number is higher than expected, as established by comparison with a randomly generated control group (Supplementary Fig. 1d) and artificially inserted boundary domains (Fig. 2b). Different domain calling algorithms and different resolutions of input Hi-C data showed similar trends (Fig. 2e-h). Gene expression levels seem somewhat irrelevant with this co-regulation (Fig. 2). Gene ontology analysis showed a high correspondence of these chromatin domain genes with immune infiltration (Fig. 3a) and the correlation of tumor

purity score with coregulated chromatin domains further strengthen this (Fig. 3b). Further, removal of these "immune domains" reduced the strength of co-regulation (Fig. 3c-e). Then the authors assess the highest "DADo" score datasets (Fig. 4a) and its top chromatin domains (Fig. 4b-e). Lastly, the author analyzed the gene family sets in the chromatin domains, and showed these co-regulated chromatin domains tend to contain more genes within the same gene family (Fig. 4f-g). The data were overall solid and interesting to wide audience.

However, the authors could have made more of an effort to show that the differentially active chromatin domains are consistent among two different cell lines. There has been evidence showing that the high order genome structure (such as topological associated domains, TADs) can be different among cancer and normal cells (Nature genetics 2020, PCAWG Consortium). Thus, using one group of Hi-C data to compare two different conditions (e.g., normal vs. cancer) might introduce errors and we were a little unclear if that was done, i.e., are co-regulated genes between two different condition cell lines within the same chromatin domain or was the domain different in cancer or normal.

The manuscript was dense but we detected at least two minor typos: On Page. 11 Line 51, it should be A549 lung cancer cell line instead of "A529". In supplementary Figure 1d, there is a typo in the legend annotation.

There are several conceptual issues that could have been covered. For example, a published analysis of chromatin domains in murine ES cells shows domains bearing active and inactive genes. The inactive genes do not display DNase I sensitivity at the proximal promoter and are thus not binding TFs that might attract nearby enhancers, yet in other cells these inactive genes are on, while the ES active gene is off. Thus, the domain could be hiding alternative gene expression programs activated during differentiation.

In conclusion, although restricted by the resolution and availability of published Hi-C data, that may cause the potential regulatory domains being screened out of the system, the authors directly showed a wide correlation between gene expression and genomic structures on a statistical level. Further studies using high resolution data such as promoter-capture Hi-C or HiChIP would likely reveal more details for the gene regulatory function of 3D genomic structures.

Reviewer #1

The authors study an interesting but subtle point relating the correspondence between contact domains in genome architecture and the co-regulation of transcription, and more generally the possible role of self-interacting chromatin domains. Here the use of a large number of Hi-C datasets is welcome.

1) The analyses that are presented seem thorough enough, even if the results are not stand-out in terms of impact. However, overall the analysis seems fairly limited in its scope and ambition, and I have a few comments regarding clarity. Some effort has gone into identifying differentially active domains, but then I would have expected to see further follow-up regarding chromatin marks, genome accessibility, chromatin compartments etc., to get a better handle on the biological events that may be happening in the most significantly co-regulated/transcribed domains. Also, the Hi-C data is only analyzed in terms of domain boundaries. I would be interesting to see if there are any interesting changes in the degree of intra- and inter-domain interaction probabilities (for example).

We thank the reviewer for the positive comments and spot on suggestions. In this revised version of the manuscript, we significantly expanded the follow up analyses on differentially active domains to address this and other concerns he/she expressed. We would like here to summarize the major revisions that we made and highlight the ones that address the points raised above:

- a) Using our recently published approach Calder (Liu et al. Nat. Comm. 2021) we mapped all chromatin domains to their respective chromatin sub-compartments (here we considered up to 8 sub-compartments). Through this analysis we found that, whereas all domains are predominantly in the A sub-compartments consistent with a greater gene density, differentially active domains are enriched in the B sub-compartments. Indeed, in the B sub-compartments we found that the average intra-domain gene correlation is higher and median mRNA expression lower than in the A sub-compartments. This finding indicates that gene co-regulation within chromatin domains is more prominent (and probably relevant) among genes with median-low expression levels. This also seems to be consistent with the hypothesis advanced by this reviewer (comment #6). All new results related to association with chromatin sub-compartments are shown in the new **Figure 5f-h**.
- b) Next, we limited the analysis on a subset of comparisons between normal and cancer samples of a given tissue, for which we had Hi-C datasets for both normal and tumor cells from the corresponding tissue. Upon matching chromatin domains identified in the compared Hi-C datasets, we verified that domain boundaries were highly conserved. However, differentially active domains switched sub-compartments more frequently than the other domains. This was evident in terms of both B-to-A or A-to-B compartment shifts (up to 3-fold increase) and overall distribution of the delta sub-compartment ranks, which measure how different is the position of a given domain in the sub-compartment hierarchies derived from different Hi-C experiments (for additional details see Liu et al. Nat. Comm. 2021). These changes in sub-compartment and compartment ranks indicate indeed that differentially activity within a domain reflects in changes of long-range inter-domain interactions. All results are shown in the new **Figure 5a-e**.
- c) Lastly, to examine intra-domain interactions in differentially active domains, we selected the specific comparison of normal vs. tumor prostate cells, for which we could retrieve matching Hi-C, mRNA, and H3K27ac ChIP-seq data for both conditions. Here, we performed a differential interactome analysis to compare local interactions within the differentially active domains and found that domains that were more active in tumor samples also comprised more significantly frequent interactions in the tumor Hi-C dataset. Similarly, domains that were more active in normal samples also comprised more significantly frequent interactions in the normal Hi-C dataset. Importantly, this significant difference in interactions was consistent with a different enrichment for H3K27ac peaks in the domain. These results indicate that differential transcriptional activity in the domain between two conditions goes along with differential H3K27ac and local interaction frequency. (Notably, these results are consistent with recent findings from our group and the Oricchio lab at EPFL – see Sungalee, Liu et al. Nat. Genetics 2021.) All these results are presented in the new **Figure 6**.

Major revisions to the main text are in red.

2) Given the high-degree of sequential (and hence intra-domain) co-localisation in homologous gene families it is important to dissect how much homology drives the occurrence of coordinated domain regulation. Similarly, the non-immune domains discussed in detail should probably not all be from homologous clusters, as they are currently (AKR and HOX families), to give more balanced examples.

This is an important point, and we would like here to clarify two things. First, we show that homology clusters are over-represented among differentially active domains (see **Figure 4a**), with more than 30% of significant domains containing

a cluster of genes representative of a single gene family compared to only ~5% of the non-significant domains. Interestingly, even when preserving genomic linear distances among genes of a same family, we found that genes belonging to a same family are more frequently found in the same domain than expected (Figure 4b). These results in fact suggest that co-regulation within chromatin domain is largely associated with homologous gene families, which, in turn, are organized in the genome not only in cluster of proximal genes, but in cluster of proximal genes within the same chromatin domain. Next, we extended the discussion of the domain presented in the lung cancer comparison to include additional domains which did not include homologous gene families. Note that this section of the manuscript is now discussed earlier in the context of differentially active domains in cancer and the corresponding figure panels have been moved to Figure 2 (Figure 2e-i).

3) Related to this, and in general, it would be better to see more exploration in the text of the biological relevance of differently active domains in multiple cell type comparisons (preferably relating to cancer specific changes rather than immunity). - In essence it would be good to have an overview of the lists presented in SuppTable 2 in terms of salient biological features, For example, it would have been helpful to see analyses relating to stratification of genes according to TF types, constitutive vs cell-type specific etc.

We agree with the reviewer and indeed we ran several type of enrichment analyses scanning for over-represented biological features in our set of differentially active domains.

In terms of biological pathway or process or TF, nothing really stood out except for the already discussed immune-related domains. However, as we explain in the manuscript, these emerge because we are here comparing tumor and normal samples from human tissue specimen exhibiting different extent of immune infiltration.

We should mention that given the diversity of conditions that we compared, we did not expect to find a strong enrichment for some specific biological processes. Indeed, we analyzed very different cancer types, subtypes, and genetic mutations, which might be driven by and depend on the activation (inactivation) of very different targets. It is also important to highlight, that although we made use of cancer mRNA datasets, the purpose of our study was not to discover new oncogenic mechanisms associated with deregulation of chromatin domains (although some of our findings indicate the potential of using our approach to do so). Our main goal was to test where, when, and to what extent gene expression is more concordant within chromatin domains than expected. The cancer mRNA datasets that we used, provided us with the unique opportunity of analyzing large sample cohorts exhibiting great transcriptional heterogeneity in diverse conditions and that were generated and analyzed in rather uniform manner (all RNA-seq TCGA data has been processed from the same unit adopting the same pipelines).

Overall, our results, significantly improved by the new analyses suggested by this reviewer, indicate that differentially active domains are enriched for

- 1) homologous gene clusters,
- 2) less efficiently transcribed genes which are frequently located in B sub-compartments,
- 3) changes long-range inter-domain interactions leading to sub-compartment repositioning, and
- 4) changes of local intra-domain interactions associated with histone acetylation.

4) Overall, it was fundamentally a good thing that immune infiltration was identified, but I feel any message becomes somewhat occluded if the cancer genes and how they relate to the differential activity, are not explored more fully.

The reviewer is correct in noting that the signal coming from “immune-domains” confounds the signal coming from co-regulation in cancer cells. We may further argue that recognizing this bias is in general important to properly interpret previous evidence indicating that gene expression is more correlated within domains than expected in cancer samples. However, the detailed exploration of potential oncogenic expression within differentially active chromatin domains would require a detailed curation of the results of each of the 58 comparisons, which as we discuss above was not the goal of this study.

Indeed, as also mentioned in response to the previous comment, we do not expect a common denominator in terms of oncogenic signal across the different comparisons, since these comparisons span heterogeneous tissues and genetic tumor features likely to exhibit similarly heterogeneous transcriptional programs. We expected however to find a common denominator in terms of structural features associated with differentially active domains and this indeed we found and highlighted in this revised version of the manuscript..

Overall, we expanded the discussion of the different domains and their potential oncogenic role within the specific comparisons highlighted in the paper, but we feel that doing the same type of in-depth analysis for all comparisons goes beyond the scope of this study.

5) Taking some biologically relevant examples, I would like to have seen some actual Hi-C maps in the most differential regions.

We have now included Hi-C maps for specific TADs that we discuss in more detail (see new **Figure 6**).

6) An idea: the authors may like to consider that 3D genome folding may have more relevance/assistance for the less efficiently transcribed, differently regulated genes via an optimization of enhancer and TF dynamics.

As already mentioned in response to the first comment of this reviewer, this was indeed a good intuition and suggestion that was supported by our results. Indeed, we found that differentially active domains were enriched in the B sub-compartments where genes showed on average lower mRNA expression but higher mRNA correlation within chromatin domains than genes and domains in the A sub-compartments (see new **Figure 5f-h**).

Some more specific points:

Around line 120. The nomenclature in the equation seems an odd; an somewhat complex mixture of textual labels and mathematical notation, which to me is not clear at first glance. For example I feel " $\log_2 FC(t)$ " is an obscure way of representing the number of genes in the region. Also, I feel "log" should only be used as a clear mathematical function.

We amended the notation to express the equation in clear mathematical terms.

Also, the textual description that accompanies the FCC formulation is unclear. What is being summed over? Regarding "Negative \log_2 " values; does this mean the selection of negative only values rather than multiplication by -1?

We have now revised this paragraph, providing a better description of the FCC formula, value interpretation, and cumulative sum curve calculation. In brief, FCC scores are equal to 1 in case of full concordance, are proximal to 0 when sign and magnitude of fold-changes exhibit little to no concordance patterns, and assume values close to -1 when the majority of genes have concordant fold-changes, but the few that are discordant exhibit significantly higher absolute fold-changes than the others (we included a **Supplementary Fig. 1a** with toy domain examples leading to very different FCC scores). Once we computed FCC scores for all domains, these scores are ranked in descending order and the cumulative sum curve of ranked FCC scores is compared to the one obtained after permuting gene-to-domain assignments (**Fig. 1b**), with permutations occurring only within the same expression quintile. The ratio of the area under the curve (AUC) defined by the observed FCC values and the AUC defined by random values can then be used to determine whether expression differences are more concordant within domains than expected (i.e., AUC ratio > 1).

Line 126: What is the size of the quantiles (decile, percentile?)

As clarified in the paragraph above, here we used quintiles, in line with a previous study where this permutation strategy was introduced (PMID: 25274727). This has been now clarified in the text.

Line 173. I think it is a bit of a stretch to say that the values are "highly" correlated.

We amended the text accordingly

Figure 1f & 1g. It is not clear in the main manuscript how the random expectation in 1g calculated and why this is so big compared to the somewhat lower values in 1f.

We apologize for the lack of clarity. The random expectation in **Fig. 1f** and **1g** is computed using the same null model, which is also the same that is used to compute the expected FCC values, i.e., random switch of gene-to-domain assignments, with permutations occurring only within the same expression quintile. The apparent discrepancy noted by the reviewer is because the percentages in **Fig. 1f** and **1g** refer to different things: in **Fig. 1f**, the Y-axis indicates to the percentage of chromatin domains with FCC = 1, in **Fig. 1g**, the Y-axis indicates the percentage of domains comprising 3 genes among all domains with FCC = 1. We clarified this in the text.

Fig2e-h The assessment of robustness is good, but could be supplemental to make room for further, more biological analyses.

We appreciate the suggestion from the reviewer. Given the numerous new biological analyses and results introduced in this revised version of the manuscript, we ended up adding two full new figures and revising the existing ones (including Fig. 2). **Fig. 2e-h** are now **Supplementary Fig 2b-e**.

Reviewer #2

One of the controversial questions in the field is whether the 3-D genomic structure regulates gene transcription. However, a systematic, comprehensive analysis of the correlation between 3D genome structures like TADs and sub-TADs, typically but not exclusively regulated by CTCF, and gene expression levels in multiple cell lines has not been performed. Zufferey and colleagues analyzed 30 different Hi-C experiments among 12 cancer cell lines to show that gene co-regulation with chromatin domains does exist albeit within a small number of domains. With further gene ontology analysis and tumor purity score analysis, the authors showed these active co-regulated domains are most strongly associated with immune response genes, suggesting these gene expression changes under the co-regulated chromatin domains is likely driven by immune infiltration. However, other interesting examples emerged. The contribution of this research is important as it provides evidence that gene expression correlates with chromatin structures. Indeed, it is our opinion that such co-regulation might be broader if finer structures were examined, as this has been reported in murine ES cells and elsewhere for specific examples.

The data in this work are rigorous. Zufferey and colleagues first introduce the methodology of "DADo" algorithm then perform it on datasets of different cell lines from TCGA to calculate the co-regulating gene sets within the same chromatin domain (Fig. 1). The number of such co-regulated domains number is higher than expected, as established by comparison with a randomly generated control group (Supplementary Fig. 1d) and artificially inserted boundary domains (Fig. 2b). Different domain calling algorithms and different resolutions of input Hi-C data showed similar trends (Fig. 2e-h). Gene expression levels seem somewhat irrelevant with this co-regulation (Fig. 2). Gene ontology analysis showed a high correspondence of these chromatin domain genes with immune infiltration (Fig. 3a) and the correlation of tumor purity score with coregulated chromatin domains further strengthen this (Fig. 3b). Further, removal of these "immune domains" reduced the strength of co-regulation (Fig. 3c-e). Then the authors assess the highest "DADo" score datasets (Fig. 4a) and its top chromatin domains (Fig. 4b-e). Lastly, the author analyzed the gene family sets in the chromatin domains, and showed these co-regulated chromatin domains tend to contain more genes within the same gene family (Fig. 4f-g). The data were overall solid and interesting to wide audience.

We really appreciate the highly positive feedback of the reviewer on the quality and rigor of our work and on its broad level of interest. As the reviewer will see, and as we summarize below, we have now included several new analyses to explore in more detail the features of differentially active domains, including detailed comparisons of Hi-C datasets matching the compared conditions as he/she suggested (see response to the next comment).

We would like here to briefly summarize all the main revisions done in this new version of the manuscript:

- a) First, we analyzed how differentially active domains are distributed across chromatin compartments and sub-compartments. Using our recently published approach Calder (Liu et al. Nat. Comm. 2021) we mapped all chromatin domains to their respective chromatin sub-compartments (here we considered up to 8 sub-compartments). We found that, whereas all domains are predominantly in the A sub-compartments consistent with a greater gene density, differentially active domains are enriched in the B sub-compartments. Indeed, in the B sub-compartments we found that the average within-domain gene correlation is higher and median mRNA expression lower than in the A sub-compartments. This finding indicates that gene co-regulation within chromatin domains is more prominent (and probably relevant) among genes with median-low expression levels. These results are shown in the new **Figure f-h**.
- b) Next, we limited the analysis on a subset of comparisons between normal and cancer samples of a given tissue, for which we had Hi-C datasets for both normal and tumor cells from the corresponding tissue. Upon matching chromatin domains identified in the compared Hi-C datasets, we verified that domain boundaries were highly conserved. However, differentially active domains switched sub-compartments more frequently than the other domains. This was evident in terms of both B-to-A or A-to-B compartment shifts (up to 3-fold increase) and overall distribution of the delta sub-compartment ranks, which measure how different is the position of a given domain in the sub-compartment hierarchies derived from different Hi-C experiments (for additional details see Liu et al. Nat. Comm. 2021). These changes in sub-compartment and compartment ranks indicate indeed that differentially activity within a domain reflects in changes of long-range inter-domain interactions. All results are shown in the new **Figure 5a-e**.

- c) Lastly, we examined intra-domain interactions in differentially active domains. We selected the specific comparison of normal vs. tumor prostate cells, for which we could retrieve matching Hi-C, mRNA, and H3K27ac ChIP-seq data for both conditions. Here, we found that domains that were more active in tumor cells also comprise more significantly frequent interactions in the tumor Hi-C dataset. Similarly, domains that were more active in normal cells also comprise more significantly frequent interactions in the normal Hi-C dataset. Importantly, this significant difference in interactions was consistent with a different enrichment for H3K27ac peaks in the domain. These results indicate that differential transcriptional activity in the domain between two conditions goes along with differential H3K27ac and local interaction frequency. (Notably, these results are consistent with recent findings from our group and the Oricchio lab at EPFL – see Sungalee, Liu et al. Nat. Genetics 2021.) These results are shown in the new **Figure 6**.

Major revisions to the main text are in red.

However, the authors could have made more of an effort to show that the differentially active chromatin domains are consistent among two different cell lines. There has been evidence showing that the high order genome structure (such as topological associated domains, TADs) can be different among cancer and normal cells (Nature genetics 2020, PCAWG Consortium). Thus, using one group of Hi-C data to compare two different conditions (e.g., normal vs. cancer) might introduce errors and we were a little unclear if that was done, i.e., are co-regulated genes between two different condition cell lines within the same chromatin domain or was the domain different in cancer or normal.

The reviewer raises here a good point, which we haven't explicitly addressed. In this revised version of the manuscript, we dedicated more space to a subset of comparisons between normal and tumor tissues for which we could compare normal and tumor cell 3D architectures (i.e., Hi-C datasets for normal and tumor cells from the same tissue). In the context of this comparison, we also compared chromatin domain boundaries detected in the two Hi-C datasets. Overall, as also observed in previous studies, we found that a large percentage of domain boundaries was preserved: 79% on average for domains that were not differentially active between the compared conditions, and 76.5% for those that were found differentially active. (Note that the global average across all comparisons is ~75%.) Although this result indicates a lower boundary conservation for significant domains, this decrease is marginal and not significant. Indeed, by randomly sampling 1000 times a number of domains equal to the number of significant domains, we found that 11% of the times the percentage of conserved boundaries was 76.5% or less (i.e., empirical p-value = 0.11). We now discuss these results in the main manuscript and in a dedicated figure panel (**Fig. 5b,c**).

The manuscript was dense, but we detected at least two minor typos: On Page. 11 Line 51, it should be A549 lung cancer cell line instead of "A529". In supplementary Figure 1d, there is a typo in the legend annotation.

We amended the text accordingly.

There are several conceptual issues that could have been covered. For example, a published analysis of chromatin domains in murine ES cells shows domains bearing active and inactive genes. The inactive genes do not display DNase I sensitivity at the proximal promoter and are thus not binding TFs that might attract nearby enhancers, yet in other cells these inactive genes are on, while the ES active gene is off. Thus, the domain could be hiding alternative gene expression programs activated during differentiation.

This is an interesting and complex point. Whereas it is difficult to explore for all comparisons, in this revised version of the manuscript, we explored in more detail matching Hi-C, mRNA, and H3K27ac ChIP-seq data for prostate cancer and normal cells. We found that gene differential activation at the domain level matched different level of H3K27ac, suggesting a different recruitment of TF and transcriptional modulators at enhancer and promoter sites. This evidence was further corroborated by different intra-domain local interactions between these regulatory elements, which were indeed more frequent when the genes were more active.

In conclusion, although restricted by the resolution and availability of published Hi-C data, that may cause the potential regulatory domains being screened out of the system, the authors directly showed a wide correlation between gene expression and genomic structures on a statistical level. Further studies using high resolution data such as promoter-capture Hi-C or HiChIP would likely reveal more details for the gene regulatory function of 3D genomic structures.

We thank the reviewer once more for his/her positive feedback. As he/she pointed out, our analyses were indeed restricted to published datasets and if on the one hand they certainly suffer from missing matching data, on the other hand they benefit from a solid statistical analysis that is possible only through multiple datasets and comparisons. Finally, we certainly agree with his/her last remarks, which we incorporated them in the discussion of this revised version of the manuscript.

Second round of review

Reviewer 1

Overall I am content with the manuscript revision and response to reviewers. The additional analyses relating to chromatin sub-types and markers help to give an impression of what is driving the differentially active structural domains. Also, the rearrangement that brings new panels to the end of Figure 2 seem to make for a better story.

I appreciate the changes to the abstract but suggest that it be tweaked a bit more in-line with the covering letter to editor, which seems more impactful in terms of biological findings. For example "differential activation of chromatin domains is not associated with major changes of domain boundaries, but rather with changes of sub-compartment and intra-domain contacts"; seems like a clear finding.

Also, in the Abstract I would change the phrase "domains of co-regulations were enriched in inactive chromatin sub-compartments" to something like "co-regulated domains were enriched in the less active B sub-compartment" or maybe to something indicating "differentially active domains switched A/B sub-compartment more frequently than the others"

Textual Points:

Line 47: I would rephrase "Interactions driven instead by histone modifications" slightly to avoid making a strong statement suggesting histone modifications are driving cause of A/B compartmental interaction. Certainly histone marks reflect the chromatin state, and can be key for the establishment of that state. However, my view is that A/B partitioning occurs because of (dis)similarity of chromatin compaction/accessibility and common regulatory factors; histone mods are required but only one aspect of this.

For Figure 2d, and its legend, I suggest using "sub-sampling" rather than "down-sampling"; to me the latter is somewhat associated with reduction of signal resolution, rather than random selection.

Figure 5c should have axis labels.

At lines 349, 354 etc. I suggest replacing the term "sub-compartment repositioning" with "sub-compartment switching". Certainly some kind of relative positional changes would happen, but what is being measured are Hi-C contact correlation changes, rather than positions per-se (as you might access with a microscope or 3D genome model).

Maybe add to the Discussion that "co-regulation might be more relevant among less efficiently transcribed genes".

In the conclusion I would specifically add that analysis of Hi-C data provides the source for the helpful, complementary information.

Authors Response

Point-by-point responses to the reviewers' comments:

Overall I am content with the manuscript revision and response to reviewers. The additional analyses relating to chromatin sub-types and markers help to give an impression of what is driving the differentially active structural domains. Also, the rearrangement that brings new panels to the end of Figure 2 seem to make for a better story.

I appreciate the changes to the abstract but suggest that it be tweaked a bit more in-line with the covering letter to editor, which seems more impactful in terms of biological findings. For example "differential activation of chromatin domains is not associated with major changes of domain boundaries, but rather with changes of sub-compartment and intra-domain contacts"; seems like a clear finding.

Also, in the Abstract I would change the phrase "domains of co-regulations were enriched in inactive chromatin sub-compartments" to something like "co-regulated domains were enriched in the less active B sub-compartment" or maybe to something indicating "differentially active domains switched A/B sub-compartment more frequently than the others"

Response: We thank the reviewer for the positive feedback to our revised work, we really appreciated his/her suggestions during the first round of review. We have now revised the abstract following his/her additional comments.

Textual Points:

Line 47: I would rephrase "Interactions driven instead by histone modifications" slightly to avoid making a strong statement suggesting histone modifications are driving cause of A/B compartmental interaction. Certainly histone marks reflect the chromatin state, and can be key for the establishment of that state. However, my view is that A/B partitioning occurs because of (dis)similarity of chromatin compaction/accessibility and common regulatory factors; histone mods are required but only one aspect of this.

For Figure 2d, and its legend, I suggest using "sub-sampling" rather than "down-sampling"; to me the latter is somewhat associated with reduction of signal resolution, rather than random selection.

Figure 5c should have axis labels.

At lines 349, 354 etc. I suggest replacing the term "sub-compartment repositioning" with "sub-compartment switching". Certainly some kind of relative positional changes would happen, but what is being measured are Hi-C contact correlation changes, rather than positions per-se (as you might access with a microscope or 3D genome model).

Maybe add to the Discussion that "co-regulation might be more relevant among less efficiently transcribed genes".

In the conclusion I would specifically add that analysis of Hi-C data provides the source for the helpful, complementary information.

Response: We have now revised the text and figures to clarify and amend terminology and concepts as suggested by the reviewer. Conversely, we recently introduced and used the term "sub-compartment repositioning" in the context of results obtained with a method we developed (see Liu et al. Nat. Comm. 2021, and Sungalee, Liu et al. Nat. Genetics 2021). Unless the reviewer expresses a strong opposition, we would prefer to keep this term for consistency with these previous studies.